# Assessment of Bioactive Profile of Sorghum Brans under the Effect of Growing Conditions and Nitrogen Fertilization

Róbert Nagy [1], Eszter Murányi [2], Piroska Bíróné Molnár [3], Judit Szepesi [1], Zoltán Győri [1], Szilvia Veres [4], Judit Remenyik [3] and Péter Sipos [1,*]

[1] Institute of Nutrition, Faculty of Agricultural and Food Sciences and Environmental Management, University of Debrecen, 138 Böszörményi Str., 4032 Debrecen, Hungary

[2] Research Institute of Karcag, Hungarian University of Agriculture and Life Sciences, 2100 Gödöllő, Hungary

[3] Institute of Food Technology, Faculty of Agricultural and Food Sciences and Environmental Management, University of Debrecen, 1 Egyetem Square, 4032 Debrecen, Hungary

[4] Department of Applied Plant Biology, Institute of Crop Sciences, Faculty of Agricultural and Food Sciences and Environmental Management, University of Debrecen, 138 Böszörményi, 4032 Debrecen, Hungary

* Correspondence: siposp@agr.unideb.hu

**Abstract:** *Sorghum bicolor* (L.) Moench is an increasingly important crop grown in many countries as a food source due to its excellent nutritional value, drought and pest resistance, and gluten-free properties. In this study, the bioactive profiles and antioxidant potentials of brans of six sorghum varieties were evaluated using spectrophotometric methods. The effects of weather and environmental conditions and different nitrogen nutrition were also evaluated. The bran of red varieties contained a higher amount of polyphenols and tannins and exhibited higher antioxidant capacities than the bran of white varieties, with the exception of one red genotype. The highest total polyphenol contents were measured in samples from two red varieties (Zádor, Alföldi1) with $1084.52 \pm 57.92$ mg $100$ g$^{-1}$ GAE and $1802.51 \pm 121.13$ mg $100$ g$^{-1}$ GAE values, respectively, while condensed tannin content varied between $0.50$ mg g$^{-1}$ and $47.79$ mg g$^{-1}$ in sorghum brans. Red varieties showed higher antioxidant activities/capacities with $70–281$ µmol TE g$^{-1}$ and $71–145$ µmol TE g$^{-1}$ for DPPH and TEAC. Correlation analysis showed a strong interaction between DPPH, TEAC, and the amounts of polyphenols and tannins, but not with FRAP values. In conclusion, red-colored varieties are a good source of polyphenols, but seed color alone is not enough to determine the nutritional value of a genotype, and the environmental conditions greatly affect the bioactive profile of sorghum.

**Keywords:** sorghum; antioxidant; polyphenols; agronomy; nutrition

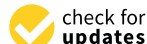



## 1. Introduction

Oxidative stress caused by reactive oxygen species (ROS) and reactive nitrogen species (RNS) has an important role both in plant and human health and vital processes [1,2]. These free radicals are present in normal metabolism and they are responsible for several cellular processes (apoptosis, signaling), but an increased amount of free radicals induce harmful effects in organisms, such as damaging cell components, lipids, DNA, and proteins [2–4]. Antioxidants such as vitamins, phenols, flavonoids, and enzymes have great importance in quenching free radicals and regulating oxidative stress by chain-breaking mechanisms or the prevention of oxidation, as well as binding metal ions responsible for oxidative damage, such as iron and copper [5–7]. Polyphenols belong to one of the major bioactive groups found in fruits and grains. Due to the large number of accessible hydroxyl groups they contain, they are one of the most important non-enzymatic antioxidant sources of plant origin [8,9]. Phenolic acids and flavonoids (flavons, flavanols, and anthocyanidins) are the major sub-groups, and aside from their antioxidant effects, they have anti-inflammatory, anti-cancer, and antibacterial effects as well. Their main role is the regulation of abiotic and biotic stress factors during plant development [10–12].

Herbs, fruits, and vegetables, such as chili or citrus fruit, are the main sources of phenols, but several grains, for example, sorghum, barley, or wheat, also contain these molecules [13]. Depending on genotype, sorghum contains higher amounts of polyphenols and flavonoids compared to other cereals, such as millets or oats. This makes sorghum a unique raw material with high nutritive value and several positive health benefits, such as the prevention of chronic diseases due to its potential to quench free radicals. It is also free from gluten, which is responsible for celiac disease and non-celiac gluten intolerance [14–19]. Despite its favorable nutritional value, it is mainly used for industrial purposes or as animal feed in Western countries, but its food utilization is increasing [20–23]. Among the numerous phenolic compounds found in sorghum, condensed tannins (CTs) and 3-deoxyanthocyanidins (3-DA) are the two most abundant and important components [10,17,19,24–26]. CTs are composed of polymerized flavan-3-ol units, predominantly (epi)catechin, (epi)gallocatechin, or (epi)gallocatechin gallate units. Depending on the degree of polymerization, we distinguish oligomeric (dimers, trimers) and polymeric CTs, which can be further divided by their monomeric units and their hydroxylation patterns (procyanidins, prodelphinidins, etc.) [24,27–29]. Procyanidin-type proanthocyanidins are the main group of tannins found in sorghum, and their presence is controlled genetically [11,19,28,30]. These tannins have a major role in protecting plants against pests and insects but also act as an anti-nutrient compound by binding to proteins and other macro- and micro-nutrients, thus decreasing the nutritional value of food matrixes. Despite this controversy, they have a significant impact on gut health, digestion, calorie intake, and the microbiome, which makes them a major addition to food products or medicine [31–33]. They are also outstanding antioxidants because of their structural characteristics and numerous functional hydroxyl groups. It was reported by several studies that CTs have an important role in maintaining intestinal health and a healthy microbiome via SCFA (short-chained fatty acids) production, inhibiting Gram-negative bacterial growth and reducing intestinal permeability [34–36].

There are also several other polyphenols, mainly flavons such as apigenin and luteolin, flavanones such as naringenin, 3-deoxyanthocyanidins such as luteolinidin and apigeninidin, and phenolic acids such as caffeic acid, cinnamic acid, and ferulic acid, which together with tannins contribute to the health benefits of sorghum grains [10,11,22,37,38]. These health benefits include a strong radical scavenging ability to protect against oxidative stress, anti-inflammatory and antibacterial effects, prevention of colon cancer, the regulation of cardiovascular diseases and kidney failure, as well as the capacity to control blood sugar level and digestion, and it can also modify and maintain the health of the human gut as a prebiotic agent for the gut microbiome [10,23,28,37,39–43].

One of the main benefits of sorghum is its tolerance against environmental factors such as drought, heat, and insect and pest damage. For this reason, sorghum is cultivated primarily as staple food mainly in semi-arid regions, such as North Africa and Australia, but also in South Asian countries, as well [14,24]. The flavonoid and polyphenol content is mainly regulated by genetic and environmental conditions, determined mostly by the color of seeds, the presence or absence of the pigmented testa, and the degree of insect and environmental damages, and can vary widely depending on growing areas [44,45]. Seed color is often a key marker during the variety-choosing process of farmers because it can indicate the polyphenol and, thus, the tannin content of cereal seeds [46,47].

Droughts, extremely high temperatures, and other environmental conditions such as soil leaching or plant diseases are increasing problems in agriculture today. There are many areas around the world where economical agricultural production has become impossible because of environmental issues and the effects of climate change [48–50]. Soil quality and composition, as well as nutrition supply, are further factors that greatly influence yields and grain quality during harvest. Nitrogen is one of the main nutrients needed for plant development and growth. The protein content is determined by the amount of nitrogen available during the growing months or specific growing periods [51–53]. There are several studies that suggest that nitrogen supply can also influence the number of other

components, such as phenols, flavonoids, and mineral content, as well as the efficient use of nitrogen during plant development. In addition, protein and polyphenol contents are interrelated, which can further determine the antioxidant and health effects of grains, fruits, and vegetables [54–59].

Due to the impact of biologically active chemical compounds in food on human health, it is important to map out the bioactive properties of food materials. The bran fraction of cereal seeds contains significant amounts of polyphenol-like compounds, which have serious health-related properties. Therefore, it can be utilized as a food additive or nutritional supplement in food matrices to improve the overall health of the population. In this study, the antioxidant properties of the total phenol and tannin content in the bran of red and white sorghum varieties from three years with different nitrogen additions were evaluated, and the effects of environmental conditions and nitrogen addition on the polyphenol and tannin content of sorghum bran were investigated.

## 2. Materials and Methods

### 2.1. Experimental Design

Field experiments were conducted on the fields of the Research Institute of Karcag (47°17′27.2″ N 20°53′27.8″ E), Hungarian University of Agriculture and Life Sciences, Hungary, in the years 2019, 2020, and 2021 in a small plot field experiment. A total of 6 sorghum varieties, 3 white and 3 red genotypes with different ripening times, were evaluated (Table 1). Two treatments were conducted in a split-plot design, one without (Control = C) and one with the addition (Treated = T) of 60 kg ha$^{-1}$ nitrogen (Péti-só, 27% N) fertilizer with a total of four replicates. The forecrops were maize in 2019 and 2020 and winter barley in 2021. Nitrogen fertilization was applied manually before sowing.

**Table 1.** Characteristics of different sorghum varieties: pericarp color, ripening time, and source of the seeds.

| Variety | Pericarp Color | Ripening Time | Source |
|---------|---------------|---------------|--------|
| Zádor | Red/brown | early | Karcag, Hungary, DE-MATE |
| Alföldi1 | Red | semi-early | Karcag, Hungary, DE-MATE |
| ES Foehn | Red | semi-early | Karcag, Hungary, DE-MATE |
| Albita | White | semi-early | Karcag, Hungary, DE-MATE |
| Albanus | White | semi-late | Karcag, Hungary, DE-MATE |
| Farmsugro 180 | White | semi-late | Karcag, Hungary, DE-MATE |

### 2.2. Properties of Experimental Site and Weather Conditions

The experimental area is located in the contact zone of Hortobágy and Nagykunság, Hungary (47°17′27.2″ N 20°53′27.8″ E). The soil of the area is plain and belongs to meadow chernozem soils, according to the genetic soil classification. Soil analysis was carried out by the Research Institute of Karcag. Samples were taken from 0–0.15 m depth diagonally in each plot. Soil data characterized the upper 15 cm of the experimental site (Table 2) as a loam–clay loam physical texture with an acidic pH level (pH = 4.5–5.4). Furthermore, the humus content of the soil was good, between 2.8 and 4%. Phosphorus and potassium contents differed greatly, depending on the sowing area. In 2019 and 2021, sorghum was grown in soil with a good phosphorus and potassium supply, while in 2020, the mineral content of the soil showed lower levels.

**Table 2.** Soil data of the experimental plots.

| Year | pH (KCl) | $K_A$ | Water Soluble Total Salt Content (m/m%) | Carbonated Lime Content (m/m%) | Humus (m/m%) | $(NO_3 + NO_2)$ N $(mg\ kg^{-1})$ | AL-$P_2O_5$ $(mg\ kg^{-1})$ | AL-$K_2O$ $(mg\ kg^{-1})$ |
|------|----------|-------|------|------|------|------|------|------|
| 2019 | 5.1 | 46 | 0.02 | <0.05 | 3.4 | 6.4 | 136 | 486 |
| 2020 | 4.7 | 39 | <0.02 | 0.21 | 2.8 | 5.2 | 87 | 255 |
| 2021 | 4.9 | 44 | <0.02 | 0.21 | 3.3 | 9.2 | 175 | 462 |

Note: $K_A$ = soil cohesion number, AL-$P_2O_5$ = Ammonium lactate soluble phosphorus–pentoxide, AL-$K_2O$ = Ammonium lactate soluble potassium–oxide. Source: Research Institute of Karcag.

Weather data from the three evaluated years are presented in Figure 1. Precipitation and temperature values were gathered by the meteorological station (VAISALA QLC–50) of the Research Institute of Karcag in 10 min intervals [60]. Average daylight duration was 249, 230, and 244 h during the three years of the experiment. In 2019, due to the large amount of precipitation in May (164.0 mm), the sowing date was postponed to the beginning of June. Most of the annual precipitation fell before maturation in winter and spring, while summer rainfall was below average, and even below the 50-year precipitation average in the area (180 mm in 2019, 217 mm long-term average). In 2020, rainfall (17.8 mm, 54 mm long-term average) and temperature were lower (14.6 °C, 16.3 °C long-term average) at the beginning of the growing period, which caused a delay in plant development. The relatively large amount of rainfall in June and July had a favorable effect on the development of plant populations. In 2021, the distribution of precipitation in winter and the growing season was also extreme. Precipitation was high in winter months, while in other months, it was below average, which resulted in the deterioration of plant development during the growing and maturing seasons. Additionally, bird damage and fungal damage caused by increased moisture levels during harvest resulted in further losses in 2021.

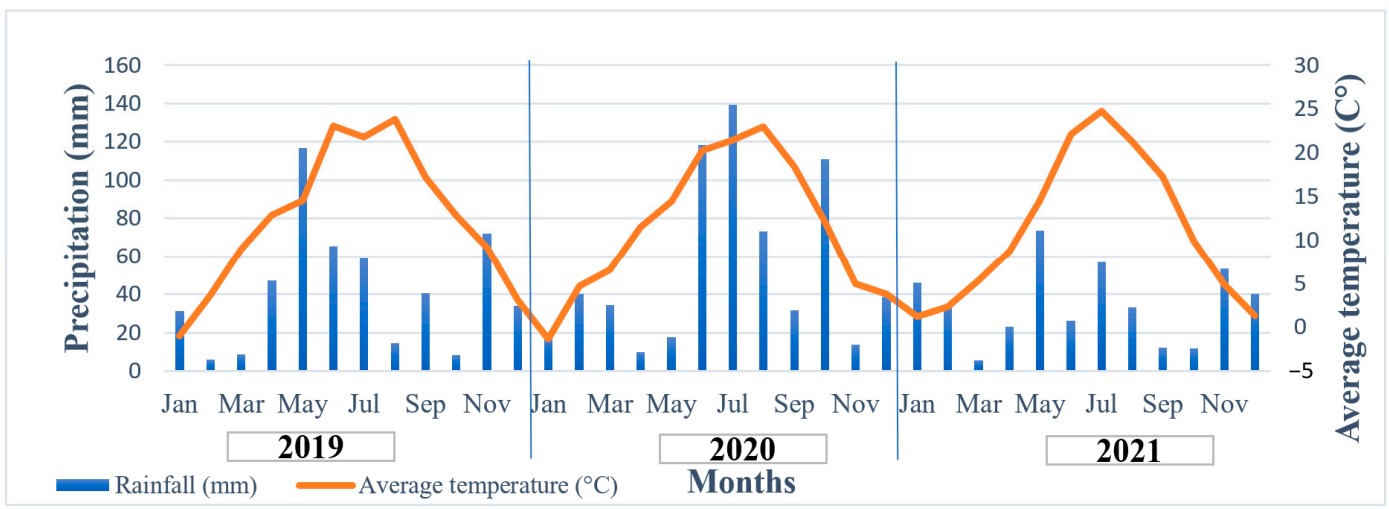

**Figure 1.** Weather conditions (precipitation in mm and monthly average temperature in °C) of the experimental years (2019, 2020, 2021). Source: Research Institute of Karcag.

### 2.3. Chemical Reagents

Folin–Ciocalteu reagent, iron–chloride (III), sodium acetate, sodium carbonate, hydrochloric acid, ascorbic acid standard, ethanol, and methanol were sourced from VWR International (Debrecen, Hungary). Gallic acid standard, ABTS (2,2′-azino-bis (3-ethylbenzothiazoline-6-sulfonic acid)), potassium persulfate, DPPH (1,1-diphenyl-2-picrylhydrazyl), vanillin, TPTZ (2,4,6-Tris(2-pyridyl)-s-triazine), and Trolox (6-hydroxy-2,5,7,8-tetramethylchroman-2-carboxylic acid) were bought from Merck (Budapest, Hungary). Catechin standard was brought from Extrasynthese (Genay, France). All chemicals and reagents used in this experiment were at least analytical grade and met quality standards.

### 2.4. Preparation of Sorghum Bran

Sorghum grains were decorticated using a laboratory SATAKE Stone Peeler at maximum output for 50 s. After homogenization by mixing, a 100 g sample was weighted on an analytical scale for every variety. Bran was separated from the kernels and used for further measurements. Samples were stored at $-20\ °C$ in airtight plastic bags prior to analysis. Measurements were taken only with the bran fractions.

### 2.5. Measurement of Total Phenolic Content

Total Phenolic Content (TPC) analysis was carried out by the Folin–Ciocalteu method, according to Singleton and Rossi, with some modifications by Nemes et al. (2018) [61,62]. After homogenization, a 0.5 g sample was weighted on an analytical scale in 4 field replicates, and a 5 mL methanol–distilled water mixture (80:20 *v/v*%) was added. The mixture was vortexed for 30 s and put into an ultrasonic water bath (25 °C, W) for 20 min. Extracts were centrifuged (Frontier 5000 Series, Ohaus Europe, Nänikon, Switzerland) at a maximum of $3600\times g$ for 10 min, and the supernatant was saved and stored at $-20\ °C$ until analysis. For the measurement, a SpectroStar nano spectrophotometer (BMG Labtech, Ortenberg, Germany) was used with a microplate reader using a TPP-96 plate. An aliquot of 10 μL extract was added to the wells, with 190 μL distilled water and 25 μL diluted Folin–Ciocalteu reagent. After 6 min, 75 μL sodium carbonate (7 *w/v*%) was added, and absorbance values were measured at 765 nm after 10 min incubation at 45 °C. Gallic acid was used as a standard, and results were given in milligrams per 100 g gallic acid equivalent (GAE). Standard curves and $R^2$ for all measurements can be seen in Figure S1. All reagents and chemicals were analytical grade.

### 2.6. Measurement of Condensed Tannins

For measuring the condensed tannin (CT) content of sorghum brans, the vanillin–HCL method was used according to Price et al. (1978) with some modifications [63]. First, 5 mL methanol was added to 0.5 g bran and vortexed for 30 s. The mixture was put into an ultrasonic water bath (25 °C, 180 W) for 20 min. Extracts were centrifuged at $3600\times g$ for 10 min, and supernatants were used for further analysis. An aliquot of 10 μL methanolic extract was put into a TPP-96 plate, and 200 μL vanillin (4 *w/v*%, Merck, Budapest, Hungary) solution was added to each well. Finally, 100 μL c.c. hydrochloric acid was added and the mixture was incubated at room temperature for 15 min. After incubation, absorbance was measured at 500 nm. Catechin was used as standard. Standard curves and $R^2$ for all measurements can be seen in Figure S2. Results were given in milligrams per gram catechin equivalent.

### 2.7. Antioxidant Content and Capacity

There are several methods with different mechanisms to measure the antioxidant content or capacities of biological and food samples. Antioxidant capacity defines the amounts of free radicals scavenged by a sample or a group of antioxidants, and it is one of the main properties used to characterize different samples. For our evaluation, 3 commonly used methods (TEAC, DPPH, and FRAP) were chosen to analyze the antioxidant capacity of sorghum brans.

Extracts prepared previously to evaluate CT content were used for TEAC and DPPH, according to the method by Zhu et al. (2009) [64] and Blois et al. (1958) [65], with some modifications by Nemes et al. (2018) [62]. ABTS (2,2′-azino-bis (3-ethylbenzothiazoline-6-sulfonic acid)) free radical and DPPH (1,1-diphenyl-2-picrylhydrazyl) free radical were used as reagents, according to the methods. A 10 μL extract was put into a TPP-96 plate and 70 μL ethanol (80 *v/v*%) was added to the wells according to the TEAC assay, while 50 μL methanol was injected for the DPPH assay. Furthermore, 190 μL TEAC reagent and DPPH reagent were added, respectively. The TEAC reagent was prepared a day prior according to regulations using ABTS radical and potassium persulfate and diluted to final volume with ethanol (80 *v/v*%). Plates were incubated at room temperature for 30 min

before measurement. Absorbance values were taken at 734 nm for TEAC and at 517 nm for DPPH assay. Trolox (6-hydroxy-2,5,7,8-tetramethylchroman-2-carboxylic acid) was used as standard; results were given as µmol Trolox equivalent per gram.

The Ferric Reduction Antioxidant Power (FRAP) assay was performed according to Benzie and Strain (1966) with some modifications by Nemes et al. (2018) [62,66]. First, 0.5 g bran was weighed on an analytical scale, and 5 mL distilled water was added. The mixture was vortexed for 30 s and put into an ultrasonic water bath (25 °C, 180 W) for 20 min. After extraction, samples were centrifuged at $3600 \times g$ for 10 min and the supernatant was saved for further analysis. An aliquot of 10 µL extract was used for spectrophotometric analysis. Samples were added to a TPP-96 plate with 30 µL distilled water and 260 µL FRAP reagent. After 8 min of incubation, absorbance values were taken at 593 nm at 37 °C. Ascorbic acid was used as standard, and results were given as µmol ascorbic acid equivalent per gram. The FRAP reagent was made on the day of analysis using TPTZ (2,4,6-Tris(2-pyridyl)-s-triazine), Iron (III) chloride, and acetate buffer. All reagents and chemicals were analytical grade. Standard curves and $R^2$ for all measurements can be seen in Figures S3–S5.

### 2.8. Statistical Analysis

A completely randomized design was adopted with sorghum brans from six varieties, with four field replicates. Data were analyzed by three-way and two-way analysis of variance, and correlation analysis was also performed using SPSS statistic software (version 24). The scatterplots and boxplots were also made by SPSS. On boxplots, circles represent the mild outliers (values that are more than one and a half time of interquartile range below Q1 or above Q3), and asterisks represent extreme outliers (values that are more than three time of interquartile range below Q1 or above Q3). Graphs and charts were made using Microsoft Excel (Microsoft Office Professional Plus 2016) and Graphpad Prism 8. All of the measurements were taken with at least 3 repetitions.

## 3. Results

### 3.1. Statistical Analysis

Variety influenced all evaluated parameters significantly ($p < 0.001$), and the same strong dependences on cropping year were proven on TPC, CTC, DPPH, and FRAP by analysis of variance. The effect of nitrogen treatment was sporadic; only the influence on TPC and FRAP was significant ($p < 0.01$). Although the influence of variety was significant in all cases and there was a significant variety × year interaction. The importance of variety seemed obvious, as the color of the kernel refers to the presence of bioactive compounds. There were no significant interactions between harvest year × treatment, and there was only one parameter (TPC), where variety x treatment interaction was significant. Variety × year × treatment interaction was significant for CTC ($p < 0.01$) and TEAC ($p < 0.05$) (Table 3).

**Table 3.** Mean square values from analysis of variance on the effect of variety, harvest year, and nitrogen treatment.

| Source | df | TPC | CTC | TEAC | DPPH | FRAP |
|---|---|---|---|---|---|---|
| Variety | 5 | 6,051,167.304 *** | 1448.58 *** | 63,561.796115 *** | 112,949.518 *** | 183.634 *** |
| Year | 2 | 398,859.462 *** | 726.757 *** | 72.362315 | 8229.524 *** | 537.477 *** |
| Treatment | 1 | 14,645.948 ** | 3.925 | 77.76431 | 39.655 | 22.091 ** |
| Variety × Year | 9 | 230,327.66 *** | 437.516 *** | 223.486019 *** | 5689.3 *** | 48.15 *** |
| Variety × Treatment | 5 | 9275.932 ** | 3.733 | 20.806931 | 97.972 | 4.192 |
| Year × Treatment | 2 | 1128.094 | 4.959 | 97.713424 | 74.734 | 5.236 |
| Variety × Year × Treatment | 9 | 3547.406 | 5.31 ** | 78.991 * | 51.474 | 2.352 |
| Error | 79 | 1970.853 | 1.881 | 37.143 | 89.566 | 1.896 |

Note: Stars after each value means statistically significant differences. * $p < 0.05$, ** $p < 0.01$, *** $p < 0.001$.

Whereas variety had a strong influence on all evaluated parameters, two–way ANOVA was applied to each variety to evaluate the effect of nitrogen fertilization and harvest years. The further figures demonstrate the averages of treatments, and the detailed results of ANOVA can be seen in Table S1.

### 3.2. Evaluation of Polyphenol Content of Sorghum Brans

The bran of evaluated varieties showed great differences in phenol content, as can be seen in Figure 2. ANOVA showed significant ($p < 0.001$) differences between harvest years for all varieties, while treatment ($p < 0.01$) and treatment × year ($p < 0.05$) had a significant effect only on some of the varieties. Among varieties, red-colored genotypes usually contained significantly higher ($p < 0.05$) amounts of total phenols, except for ES Foehn, which was similar to white genotypes. Alföldi1 had the highest amount of polyphenols, with 945–1802 mg 100 g$^{-1}$ GAE, compared to white varieties, which contained around 70–170 mg 100 g$^{-1}$ GAE. The effect of nitrogen fertilization on total phenol content was different. Most of the studied genotypes did not show any difference in TPC content under the effect of fertilization. In the third year, two genotypes, Albita and Farmsugro 180, showed greater differences as an effect of the treatments, with $94.56 \pm 8.45$, $121.92 \pm 9.34$ mg 100 g$^{-1}$ and $131.93 \pm 34.19$, $174.10 \pm 14.99$ mg 100 g$^{-1}$, respectively.

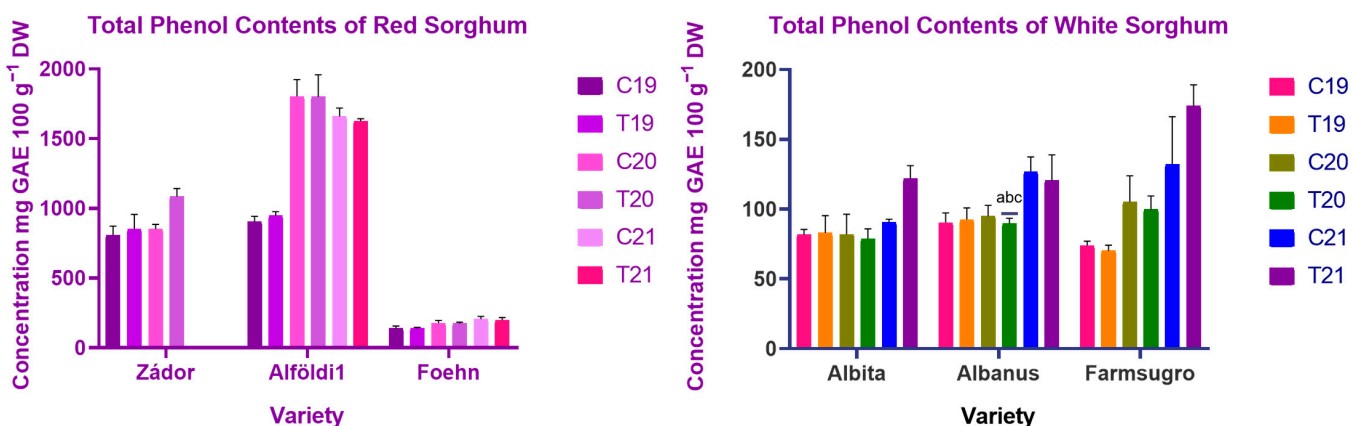

**Figure 2.** Total phenol content of red and white sorghum varieties. Note: C = Control, T = Treated, DW = Dry weight.

Regarding the effect of weather and other environmental conditions, there were significant differences between the third year and the previous two years. The third year was droughty and warmer than the previous ones, especially during the growing season. This impacted the total phenol content of sorghum brans, except for Alföldi1, which had a lower amount of total phenols compared to the previous samples; however, these differences were not statistically proven.

### 3.3. Estimation of Condensed Tannin Content and Differences of Brans

Since the number of tannins fundamentally determines the utilization of sorghum, it is extremely important to obtain information about the factors influencing the tannin content of sorghum. The composition is strongly influenced by breeding and genetic factors; for this reason, differences between varieties were expected during our study. Harvest years with different environmental conditions had a significant ($p < 0.001$) effect on tannin contents for each variety, while treatment had no proven effect. Treatment x year had a significant effect only for 1 variety ($p < 0.05$). Red varieties tended to have a higher level of CTs except for one genotype, ES Foehn, which exhibited values similar to the ones found in white genotypes in the case of total phenol content. Alföldi1 had the highest amount of condensed tannins in all years with 7.90–47.49 mg g$^{-1}$, followed by Zádor with 5.65–8.81 mg g$^{-1}$ tannin content,

respectively (Figure 3). White genotypes and ES Foehn contained a very low amount of tannins with a maximum of 3.95 mg g$^{-1}$.

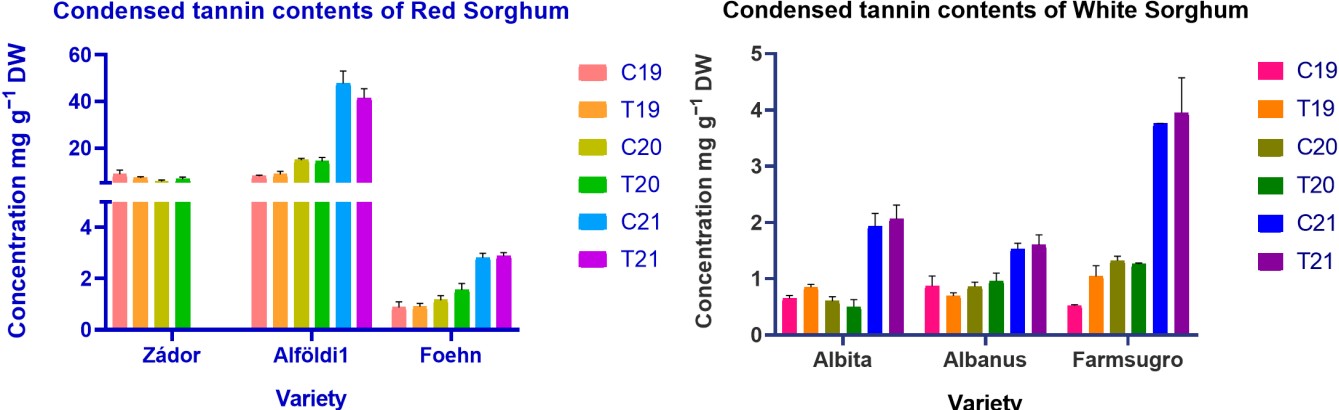

**Figure 3.** Condensed tannin content red and white sorghum varieties. Note: C = Control, T = Treated, DW = Dry weight.

Weather and environmental conditions had considerable effects on CT content in all varieties. Due to the similar weather conditions in 2019 and 2020, CT content was also found to be similar in all genotypes. However, a significantly increased TC level was detected in the 2021 samples as the result of much drier and hotter weather during the plant growing and seed maturation periods. Another factor influencing the number of tannins was the increased presence of pests and insects in 2021, which caused larger damage to the grain and a higher amount of CTs, even in white varieties, which are basically tannin-free genotypes.

*3.4. Antioxidant Properties of Sorghum Brans*

The bran fraction of a seed contains several components (phenolic acids, flavones, flavonols, flavanones, and condensed tannins) which have an impact on human health with the ability to scavenge free radicals or prevent disease. The radical scavenging and quenching ability of these compounds are dependent on various factors such as general structure, number and position of hydroxyl and ketone groups, and the size of the molecules. Measured antioxidant capacity values can be seen in Figures 4–6.

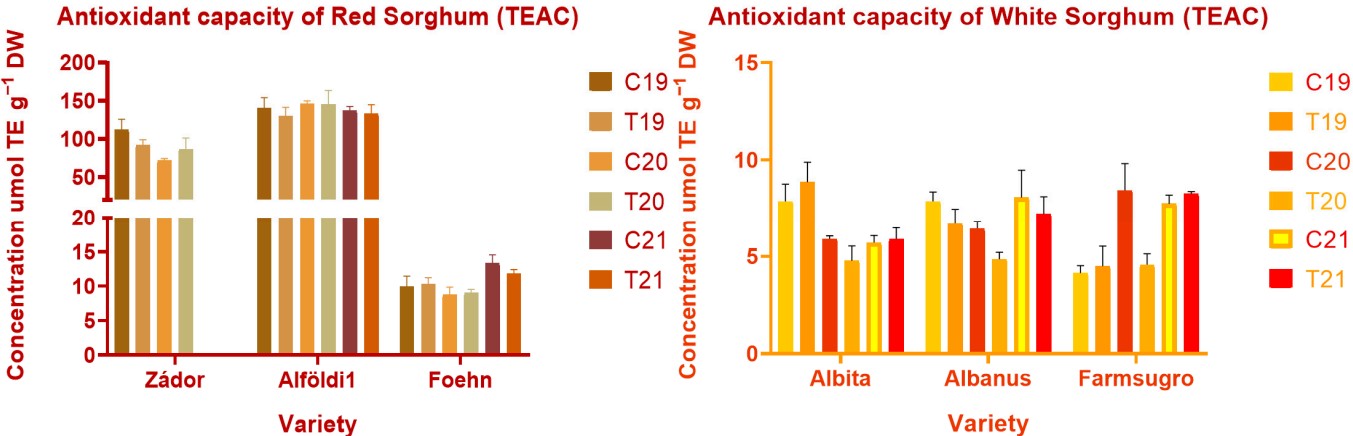

**Figure 4.** Antioxidant capacity of red and white sorghum varieties measured by TEAC assay. Note: C = Control, T = Treated, DW = Dry weight.

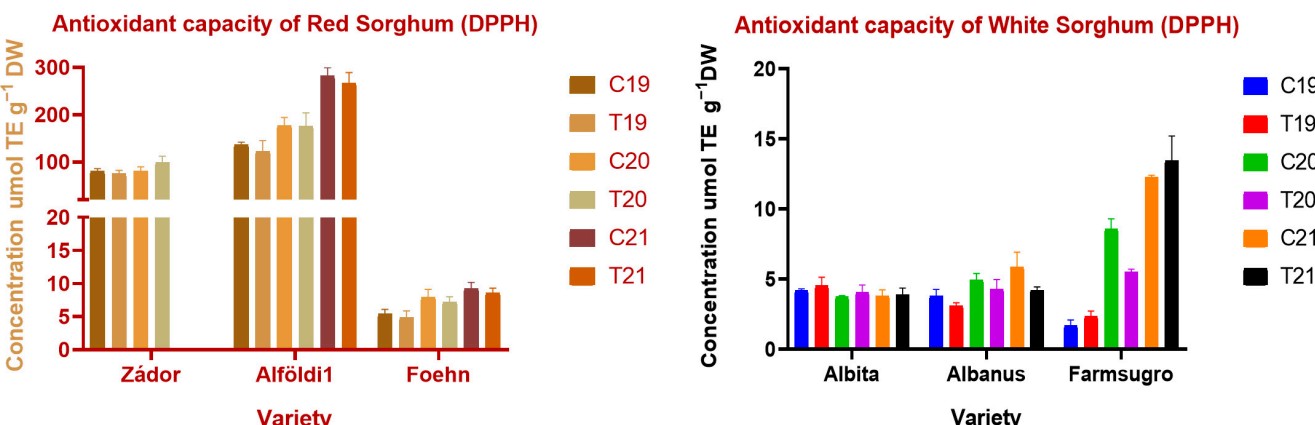

**Figure 5.** Antioxidant capacity of red and white sorghum varieties measured by DPPH assay Note: C = Control, T = Treated, DW = Dry weight.

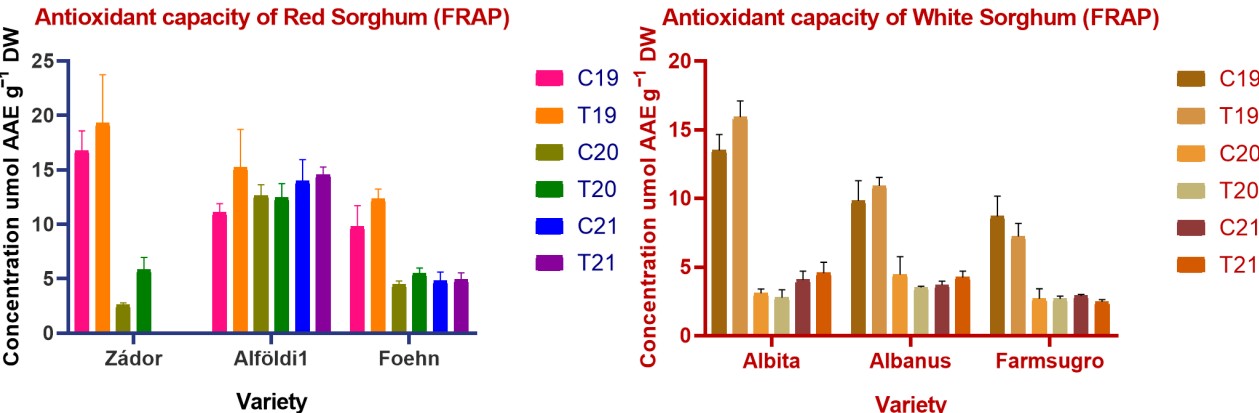

**Figure 6.** Antioxidant capacity of red and white sorghum varieties measured by FRAP. Note: C = Control, T = Treated, DW = Dry weight.

In the case of Trolox, equivalent antioxidant capacity environmental conditions had significant ($p < 0.001$, $p < 0.01$) effects on all varieties except for Alföldi1. Treatment and treatment $\times$ year interaction had no significant effect on TEAC values. Similarly to previous findings, red varieties showed significantly higher antioxidant potential than white genotypes, with the exception of ES Foehn again. Alföldi1 had the highest amount of antioxidant capacity with $122.5 \pm 19.35$–$281.7 \pm 17.57$ µmol TE g$^{-1}$ DW values in TEAC and DPPH assays. White varieties showed low values of antioxidant capacity, which correlates with their low TPC and CT contents. The FRAP values differed greatly from the results obtained by other methods. The FRAP results indicated much lower levels of antioxidant capacities, showing that there are no significant levels of water-extractable antioxidants in sorghum grains.

Nitrogen fertilization had a slight effect on antioxidant properties, which was not consistent, and there is no clear evidence that it is caused by nitrogen addition. ANOVA proved a significant effect ($p < 0.01$) in the case of Albanus. However, differences were found between different years. Effects were significant ($p < 0.01$) for all varieties except Zádor and Albita. TEAC and DPPH values were higher in red varieties in 2021, which can be explained by the higher amount of polyphenols or CT content. White varieties had a lower level of antioxidants in all years, but more significant differences between varieties and treatments were found among them. The fertilized Albita had a higher TEAC value than the control sample in 2019; however, the value was lower in the fertilized sample in 2020, but no differences in TEAC were found between the two in 2021. Farmsugro showed higher amounts of antioxidants in the control sample of 2020. Red varieties were more

heterogeneous than white varieties in terms of DPPH values, with a 30-fold difference between Alföldi1 and ES Foehn, which was interconnected with measured tannin and total phenol values.

FRAP values showed similar tendencies to other measured antioxidant values in terms of statistical differences and effects. Harvest year had a clear effect on FRAP values with significant ($p < 0.001$) differences. Meanwhile, treatment and treatment × year had no effect on the antioxidant values measured by FRAP.

### 3.5. Connections between the Antioxidant Properties, Tannin, and Polyphenol Content of Sorghum Grains

Pearson correlation analysis was performed to determine any correlation between the antioxidant potential and the number of phenols and tannins in the brans of different genotypes. Correlation coefficients are presented in Table 4.

**Table 4.** Correlation analysis of antioxidant properties, polyphenol, and tannin content.

| | Correlations | | | | |
|---|---|---|---|---|---|
| | **TPC** | **CTC** | **TEAC** | **DPPH** | **FRAP** |
| TPC | 1 | 0.798 ** | 0.963 ** | 0.947 ** | 0.561 ** |
| CTC | | 1 | 0.738 ** | 0.916 ** | 0.440 ** |
| TEAC | | | 1 | 0.925 ** | 0.628 ** |
| DPPH | | | | 1 | 0.558 ** |
| FRAP | | | | | 1 |

Notes: ** Correlation is significant at the 0.01 level (2-tailed).

Phenols are an exceedingly diverse group with many subgroups and compounds, and they are one of the most important components of sorghum, responsible for many of its beneficial health effects. There are many subgroups of phenols with different structures (hydroxyl groups, unsaturated bonds, etc.), which affects their antioxidant potential to a large extent and contribute to their antioxidant capacities. Genotypes and environmental conditions that occur in the year of harvest can also influence the amount and type of secondary metabolites synthesized during maturation. The correlations found in our research can be seen in Figure 7.

According to the Pearson correlation analysis, total phenol and condensed tannin content have a significant ($p < 0.01$) correlation. The high correlation coefficient (0.798) indicates that condensed tannins are a huge part of the total phenols in sorghum grains. However, genotype and environmental factors have a significant effect on this correlation. White varieties had low amounts of polyphenolic components and generally also had a very low level of tannins, while red genotypes with a high level of polyphenols showed significant differences between different years and varieties. Alföldi1 had a significant spike in 2021 in tannin content, while TPC had a slight decrease compared to the previous year, which was caused by the differences in weather conditions.

Total phenol content showed a significant ($p < 0.01$) correlation with antioxidant capacities for all three methods, with coefficients of 0.963, 0.947, and 0.561 for TEAC, DPPH, and FRAP, respectively. This connection appears to be stronger for TEAC and DPPH than for FRAP. These differences are caused by the different working mechanisms of measurement methods. The TEAC and DPPH are methods that measure the electron and H+ donation ability of antioxidants, while FRAP is based on ferric-reducing ability. Based on the correlation coefficients between antioxidant capacities and the measured polyphenol content, TEAC and DPPH methods are more suitable for antioxidant measurements of sorghum polyphenols, but there is a significant number of secondary metabolites with ferric reducing abilities as well present in sorghum brans, which requires further investigations. In the case of TEAC and DPPH, the correlation between TPC and TEAC/DDPH have

similar tendencies to TPC–CTC values, where red genotypes (with the exception of Foehn) are significantly different from white varieties, and there are substantial discrepancies between harvest years as well.

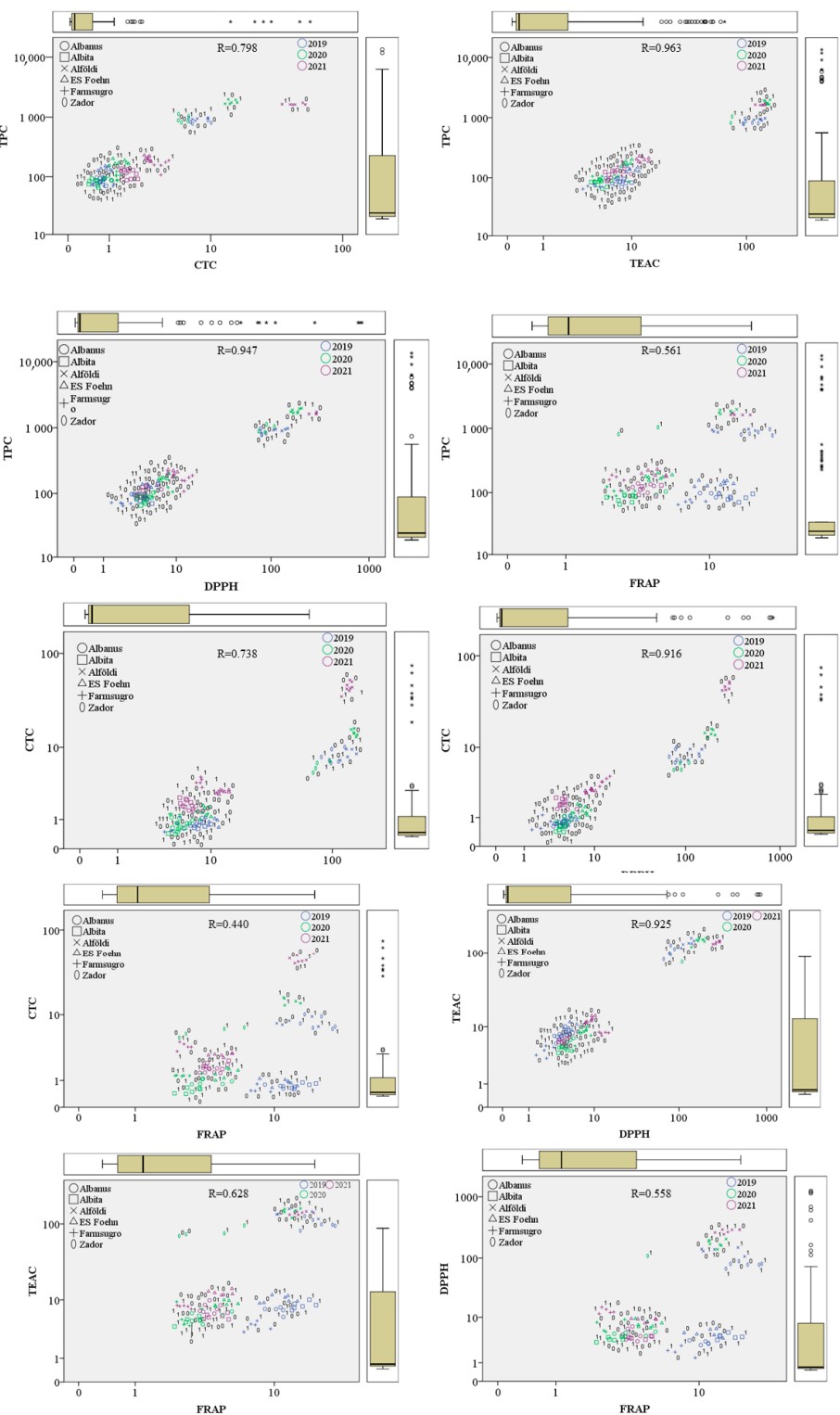

**Figure 7.** Scatter plots of the relationship between phenolic components of red and white sorghum brans and their antioxidant activities. Antioxidant values are presented as μmol g$^{-1}$ DW, TPC as mg 100 g$^{-1}$ DW and CT as mg g$^{-1}$ DW. Circles and asterisks indicate mild and extreme outliers, respectively.

Condensed tannins are extraordinary antioxidants since they have numerous accessible hydroxyl groups. This was evident from the correlation coefficients of CTC and antioxidant capacity values, which showed a significant ($p < 0.01$) interaction between the two (with 0.738 and 0.916 correlation coefficients for TEAC and DPPH, respectively). The correlation between tannin content and antioxidant capacities measured by the FRAP method was slightly less pronounced but still more significant than in the case of the previous two methods. In our study, DPPH measurement was more suitable for measuring the antioxidant potential of polyphenols, especially tannins of sorghum brans. This can be explained by the high amount of hydroxyl groups, which can act either as electron or hydrogen donors. Red varieties, with the exception of ES Foehn, had significantly higher antioxidant capacities than white varieties, but while TEAC values did not differ with increased tannin content, DPPH values spiked in the third year of the experiment with a higher CTC content.

Scatter plots were made for all three antioxidant capacity measurements as well to verify the connections between different parameters and evaluate the differences in working mechanisms. The TEAC and DPPH methods are similar to each other in terms of working mechanisms, as both are based on electron transport. This was shown in their correlation coefficients of 0.925 as well. In contrast, the FRAP method works based on the evaluation of the ability to reduce ferric ions. Both DPPH–FRAP and TEAC–FRAP had lower correlation coefficients (0.558 and 0.628, respectively).

## 4. Discussion

The aim of this study was to determine the nutraceutical value of six sorghum varieties grouped by seed color and to evaluate the effect of nitrogen fertilization and environmental conditions on the polyphenol and tannin content and antioxidant properties of grains. Genotypes harvested in three years were compared with or without nitrogen addition, measuring their total phenol and condensed tannin content, as well as their antioxidant properties.

Seed color was found to be a good indicator in the prediction of the polyphenol composition of sorghum [67,68]. Anthocyanins, flavonols, and tannins are the main compounds responsible for color development in plant seeds, and they accumulate mainly in the pigmented testa layer. Genotypes without this layer tend to have lower amounts of polyphenols, and they are brighter in color. In general, white-colored varieties have lower (11–370 mg 100 g$^{-1}$ GAE) amounts of phenols, while colored genotypes (red, brown, black) have higher amounts (480–2200 mg 100 g$^{-1}$ GAE) [37,69–74]. In our research, red genotypes showed considerable deviations between varieties, ranging from an equivalent of 200 to 1800 mg 100 g$^{-1}$ GAE, while white varieties contained low levels of phenolic compounds with 70–170 mg 100 g$^{-1}$ GAE values. These results are in line with the findings reported in the literature [37,74]. These amounts are mainly controlled by line-specific genes, but there were differences between growing years with different environmental conditions. Due to pest damage and drought in 2020 and especially in 2021, most of the varieties showed an increased level of total phenol content. Pinheiro found similar trends in terms of the flavonoid content of tannin-free sorghum varieties [75]. These findings indicate that suboptimal environmental conditions can induce synthetic pathways to produce phenol-type compounds to protect against biotic and abiotic stress factors, but in our study, no decisive proof was found; only white genotypes showed significant increases in total phenols in the 2020 and 2021 samples.

Some earlier research found a relationship between nutrition supply and secondary metabolite synthesis, suggesting a delicate balance between them. In our research, nitrogen addition at 60 kg ha$^{-1}$ did not have a clear effect on the total phenol content in comparison to an unfertilized control sample; only two varieties (Zádor, Albita) showed a significant ($p < 0.05$) increase in phenols, while others did not exhibit any changes.

Among the many flavonoids, condensed tannins are one of the most important subgroups, found in sorghum in an amount higher than in some other tannin-containing cereals such as red wheat or barley [37,76,77]. CTs content mainly depends on the presence

of the pigmented testa, and they are key components in determining the color of sorghum grain. Varieties without this testa layer, such as the white ones, usually have a low amount of CTs, whereas red, brown, and black ones contain CTs in a wide range. Condensed tannin content ranges from 1.26 to 35 mg g$^{-1}$ catechin equivalent in different types of sorghum brans [69,78,79]. Our findings confirmed that colored, darker seeds contain a higher amount of tannins, but there are some influencing factors, such as breeding and genetic properties, which can modify the tannin content of sorghum grains. Aside from the genetic-related factors, environmental conditions are the main parameters influencing CT content. Yields harvested with increased drought damage had elevated levels of tannins in the bran fraction of the kernels, even in varieties with a very low level of tannins.

Nitrogen treatment had an inconsistent effect on the tannin content in some genotypes, such as Alföldi and Foehn, whereas no differences were found in other varieties, such as Albanus and Albita. Consequently, it can be assumed that no definite connection between nitrogen fertilization and the condensed tannin content of sorghum was detected in this study.

Phenols and flavonoids have strong antioxidant properties, and they play an important role in free-radical scavenging and oxidative stress reduction. Flavonoids are particularly effective scavengers because of their multiple hydroxyl groups, double bonds, and structural complexity. Sorghum grains contain a wide array of flavonoids, and, according to the literature, sorghum antioxidant capacity values can reach levels as high as 720 and 800 μmol g$^{-1}$ Trolox or Ascorbic acid equivalent, depending on genotypes [70–72].

Antioxidant powers differ greatly depending on the method used because of differences in the working mechanism of the methods. There were significant differences between TEAC and DPPH values in the case of Alföldi1, while other varieties showed similar correlation coefficients with both methods. This difference could be caused by the increased CT content of Alföldi1 in 2021.

Total phenol content and polyphenol composition are key parameters of antioxidant power, and thus, genotype and environmental conditions have a huge impact on defending against oxidative stress factors. Nitrogen addition does not result in any significant changes in the antioxidant capacities as well.

Different antioxidant capacity assays have different mechanisms for measuring the radical scavenging and quenching ability of antioxidants. As a result, depending on which method is used, different antioxidant values can be measured. Methods capable of measuring antioxidants via both SET and HAT mechanisms (TEAC, DPPH) are more efficient for sorghum samples. There were significant differences between TPC, CTC, and TEAC, DPPH interrelations. The total phenol content of sorghum genotypes showed similar correlations for both TEAC and DPPH assays, so it can be assumed that phenols are the focal antioxidants of sorghum. However, condensed tannin content had a weaker relationship with the TEAC assay than with the DPPH assay, and the correlation between CTC and DPPH was found to be similar to the correlation between TPC and CTC. According to these results, it can be assumed that the measured DPPH values are related to the condensed tannin content of sorghum to a large degree.

FRAP values had a fairly low but significant correlation for all other variables, and it can be assumed that there are some other non-phenol-type components with antioxidant potential in sorghum.

Because of the harmful effects of changing climate conditions and soil degradation due to intense agricultural activity, it will become increasingly important to adapt to using alternative crops and plants for food production, especially ones resistant to unfavorable growing conditions. Sorghum has the potential to become the leading gluten-free crop in the Western diet and agriculture due to its excellent agronomic sustainability and high nutritional value as an antioxidant and nutraceutical source. Several countries have already discovered the advantages this cereal can provide. For this reason, it is increasingly important to get to know the sorghum varieties already in production and evaluate their possible utilization opportunities as functional ingredients in the food industry.

## 5. Conclusions

In this study, several sorghum brans were evaluated for their polyphenol, CT content, and antioxidant properties, while the influence of environmental and agricultural factors was also analyzed. Generally, red varieties were shown to be richer in polyphenol compounds, but not exclusively, as there was one red variety with low polyphenol and tannin content. Extreme weather conditions and the presence of harmful organisms increased the defensive response of plants, causing a significantly higher concentration of tannins in the brans of some red sorghum varieties. However, the nitrogen supply used in the current experiment had no significant effect on the bioactive profile of sorghum seeds. Correlation analysis showed that CTs contribute a large proportion to the antioxidant effect of sorghum grains, and their amount is crucial both in agriculture and human health and diet. Overall, sorghum has excellent biological value with good tolerance against worsening environmental conditions, and it can become an outstanding substitute for maize as a gluten-free cereal in dry areas.

**Supplementary Materials:** The following supporting information can be downloaded at: https://www.mdpi.com/article/10.3390/agriculture13040760/s1, Figure S1. Standard calibration curves of total phenol assays 2019–2021; Figure S2. Standard calibration curves of condensed tannin assays 2019–2021; Figure S3. Standard calibration curves of TEAC 2019–2021; Figure S4. Standard calibration curves of DPPH 2019–2021; Figure S5. Standard calibration curves of FRAP 2019–2021; Table S1. Mean square values from analysis of variance on the effect of harvest year and nitrogen treatment.

**Author Contributions:** Conceptualization, R.N. and P.S.; methodology, J.R. and P.B.M.; formal analysis, P.S. and J.S.; resources, S.V., E.M., J.R. and Z.G.; writing—original draft preparation, R.N. and E.M.; writing—review and editing, P.S., S.V. and J.S. visualization, R.N.; supervision, P.S. and J.R.; funding acquisition, Z.G. All authors have read and agreed to the published version of the manuscript.

**Funding:** This research received no external funding.

**Institutional Review Board Statement:** Not applicable.

**Data Availability Statement:** The data that support the findings of this study are available from the corresponding author (P.S.), upon reasonable request.

**Acknowledgments:** The authors would like to thank the Department of Applied Plant Biology, Faculty of Agricultural and Food Sciences, and Environmental Management, University of Debrecen, and the Research Institute of Karcag, Hungarian University of Agriculture and Life Sciences for providing samples for the research.

**Conflicts of Interest:** The authors declare no conflict of interest.

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
