# Peer review of "Assessment of Bioactive Profile of Sorghum Brans under the Effect of Growing Conditions and Nitrogen Fertilization"

_agriculture, doi:10.3390/agriculture13040760_

Round 1
Reviewer 1 Report
The manuscript entitled: 'Evaluation of the bioactive profile of sorghum bran under the influence of cultivation conditions and nitrogen fertilization' was evaluated. The manuscript describes that Sorghum bicolor (L.) Moench is an increasingly important plant cultivated in many countries. It is used in many industries, including: as a source of food due to its excellent nutritional value, resistance to drought and pests, and lack of gluten. In this study, the profile of bioactive compounds and antioxidant potential of six varieties of sorghum was presented. Spectrophotometric methods were used for this purpose. Observations on the effects of weather and environment, and the conditions of nitrogen fertilization added more value to the work. The chapter is well described in the manuscript - introduction, material and methods - these chapters are very readable and the literature is well selected. The obtained results are presented in the chapter description of the results. Some of them are presented in charts and tables, respectively. it is clear and legible. During the research, it was found that the bran of red varieties contained a greater amount of polyphenols, tannins and showed greater antioxidant capacity than white varieties, with the exception of one red genotype. The highest total polyphenols content measured in samples from two red varieties was 1084.52 ± 57.92 mg 100 g-1 GAE and 1802.51 ± 121.13 mg 100 g-1 of GAE value, respectively, while the content of condensed tannins ranged from 0, 50 mg g-1 to 47.79 mg g-1 in sorghum bran. Red varieties showed higher antioxidant activity. Well-chosen status analysis - Correlation analysis showed a strong interaction between DPPH, TEAC and the amount of polyphenols and tannins, but not with FRAP values. In conclusion, red varieties are a good source of polyphenols, but the color of the seeds alone is not enough to determine the nutritional value of the genotype, and the environment, climatic conditions greatly affect the bioactive profile of sorghum. Extreme weather conditions and the presence of harmful organisms increased the plant's defense response, resulting in a much higher concentration of tannins in the bran of some varieties of red sorghum. However, the nitrogen feed used in the present experiment had no significant effect on the bioactive profile of sorghum seeds. Overall, sorghum has excellent biological value with good tolerance to deteriorating environmental conditions and can become an excellent substitute for gluten-free cereals. to sum up: the manuscript is correct, legible, brings to the current science a lot of interesting information that can be used in your research and cultivation. The tables and graphs are correct. The research methods are well described, and the research itself is well supported by the literature, but I would like to suggest improving my manuscript with the latest literature in this field
I recomend more references, f.eg.:
Przybylska-Balcerek, Anna, Jakub Frankowski, and Kinga Stuper-Szablewska. "Bioactive compounds in sorghum." European Food Research and Technology 245 (2019): 1075-1080.
Frankowski, J., Przybylska-Balcerek, A., & Stuper-Szablewska, K. (2022). Concentration of Pro-Health Compound of Sorghum Grain-Based Foods. Foods, 11(2), 216.
Przybylska-Balcerek, A., Frankowski, J., & Stuper-Szablewska, K. (2020). The influence of weather conditions on bioactive compound content in sorghum grain. European Food Research and Technology, 246, 13-22.
Author Response
Thank you for your insight and recommendations and found our paper a correct and legible manuscript, which brings to the current science a lot of interesting information that can be used in research and cultivation. The aforementioned literatures were incorporated and cited in the introduction section.
Reviewer 2 Report
The nutraceutical value of sorghum depends on the genotype; however, growth-related factors may affect the bioactive content of grains and consequently their potential antioxidant activities. The authors conducted a series of experiments for this purpose. In this study, they examined the bioactive profile of sorghum brans under the effect of growing conditions and nitrogen fertilization and obtained some interesting results.
However, the authors did not report whether the interaction between genotype, growth conditions, and N fertilization (genotype x environment x N fertilization) was significant or not for all nutraceutical traits evaluated in this manuscript. Therefore I have some suggestions aimed at improving the paper.
Introduction:
I advise the authors to cite this review as it summarised the health benefit of sorghum compounds. “de Morais Cardoso, L., Pinheiro, S. S., Martino, H. S. D., & Pinheiro-Sant'Ana, H. M. (2017). Sorghum (Sorghum bicolor L.): Nutrients, bioactive compounds, and potential impact on human health. Critical reviews in food science and nutrition, 57(2), 372-390.”
Line 71-72: “ …but also… acting as anti-nutrients (or anti-nutritional factors)”. So please, rephrase.
Materials and Methods
I suggest to add the average of daylight duration (the average of photoperiod)
Statistical analysis
In this paper the authors examine the effect of more than 1 factor (genotype, growing conditions and nitrogen fertilization) on nutraceutical value of sorghum. Therefore, at least 2-way ANOVA analysis will be appropriate. Indeed, a one-way ANOVA uses one independent variable, while a two-way ANOVA uses two independent variables.
Results
Please show (as mentioned previously) in a table the interaction between genotype x growing conditions x nitrogen fertilization for all traits assessed in this paper.
For example, see the Table 2 of reference 45 cited in this paper (Line 598-599).
Author Response
We would like to thank you for your advices and time for reviewing this paper. We think that your suggestions improved the scientific value of the manuscript. Our notes are added after the paragraphs of review typed in red.
The nutraceutical value of sorghum depends on the genotype; however, growth-related factors may affect the bioactive content of grains and consequently their potential antioxidant activities. The authors conducted a series of experiments for this purpose. In this study, they examined the bioactive profile of sorghum brans under the effect of growing conditions and nitrogen fertilization and obtained some interesting results.
However, the authors did not report whether the interaction between genotype, growth conditions, and N fertilization (genotype x environment x N fertilization) was significant or not for all nutraceutical traits evaluated in this manuscript. Therefore I have some suggestions aimed at improving the paper.
Introduction:
I advise the authors to cite this review as it summarised the health benefit of sorghum compounds. “de Morais Cardoso, L., Pinheiro, S. S., Martino, H. S. D., & Pinheiro-Sant'Ana, H. M. (2017). Sorghum (Sorghum bicolor L.): Nutrients, bioactive compounds, and potential impact on human health. Critical reviews in food science and nutrition, 57(2), 372-390.”
Line 71-72: “ …but also… acting as anti-nutrients (or anti-nutritional factors)”. So please, rephrase.
The recommended paper was cited in the introduction. Text was rephrased as asked.
Materials and Methods
I suggest to add the average of daylight duration (the average of photoperiod)
Thank you for your suggestion, average daylight durations for all experimental years were added to the chapter.
Statistical analysis
In this paper the authors examine the effect of more than 1 factor (genotype, growing conditions and nitrogen fertilization) on nutraceutical value of sorghum. Therefore, at least 2-way ANOVA analysis will be appropriate. Indeed, a one-way ANOVA uses one independent variable, while a two-way ANOVA uses two independent variables.
Thank you for pointing this out. We modified the statistical analysis accordingly, and applied a three–way ANOVA to evaluate the effects of variety, year and treatment for all parameters, followed by a two–way ANOVA to evaluate the effects of year and treatment by varieties.
Results
Please show (as mentioned previously) in a table the interaction between genotype x growing conditions x nitrogen fertilization for all traits assessed in this paper.
For example, see the Table 2 of reference 45 cited in this paper (Line 598-599).
A new subchapter was added for this statistical analysis. It was done according the cited paper and its result has been summarized in Table 3. Later, 2–way ANOVA was used to evaluate the effects of year and treatment and their interactions interactions.
Reviewer 3 Report
In the present study, the bioactive profiles and antioxidant potentials of brans of six sorghum varieties were evaluated in relation different growing condition and nitrogen fertilization regimes.
Lines 25-26: the varieties have to be indicated in order to correspond its polyphenol content value to specific cultivars.
Lines 30-32: the authors mention in the conclusions the effect of environmental conditions on bran polyphenols content, but there is no such reference in the abstract where only the effect of the color of bran is highlighted.
Introduction should focus on the effect of environmental conditions and growing practices on the chemical composition of sorghum. Figures 1 and 2 are redundant and could be more appropriate in areview paper.
Generally, put the reference number before and not after the points (.).
The authors performed an one-way ANOVA although they conclude differences in terms of colour and they also present data from different years. I suggest to either perform a one way ANOVA with all the cultivars at the same time for different years or a two way ANOVA testing the cultivars and the years.
Author Response
We would like to thank you for your advices and time for reviewing this paper. We think that your suggestions improved the scientific value of the manuscript. Our notes are added after the paragraphs of review typed in red.
In the present study, the bioactive profiles and antioxidant potentials of brans of six sorghum varieties were evaluated in relation different growing condition and nitrogen fertilization regimes.
Lines 25-26: the varieties have to be indicated in order to correspond its polyphenol content value to specific cultivars.
Lines 30-32: the authors mention in the conclusions the effect of environmental conditions on bran polyphenols content, but there is no such reference in the abstract where only the effect of the color of bran is highlighted.
Introduction should focus on the effect of environmental conditions and growing practices on the chemical composition of sorghum. Figures 1 and 2 are redundant and could be more appropriate in areview paper.
Generally, put the reference number before and not after the points (.).
Thank you for your advices. All the aforementioned mistakes were corrected, Figure 1 and 2 were removed from introduction.
The authors performed an one-way ANOVA although they conclude differences in terms of colour and they also present data from different years. I suggest to either perform a one way ANOVA with all the cultivars at the same time for different years or a two way ANOVA testing the cultivars and the years.
Thank you for pointing this out. We modified the statistical analysis and applied a three–way ANOVA to evaluate the effects of variety, year and treatment for all parameters, followed by a two–way ANOVA to evaluate the effects of year and treatment by varieties.
Reviewer 4 Report
This paper describes the assessment of bioactive profile of sorghum brans under the effect of growing conditions and nitrogen fertilization. The article is quite complete, it is of interest to the scientific community, the methods and statistics used are appropriate and the results and discussion are conveniently described. The work is well discussed and is supported by the references provided by the authors. On the other hand, I consider that the Materials and Methods” section should be described in greater depth and scientific rigor. The work is interesting and delves in the knowledge of bioactive profile of different kinds of sorghum brans.
I consider that the article is appropriate to be published in Agriculture journal once the authors have made major modifications to it.
Major aspects:
Material and Methods: I recommend the authors to include a section of "reagents".
Material and Methods: Lines 164, 174, 176, 177, 231, …: Include the city and country of all the companies cited, and cite the companies of all the reagents and equipment’s employed. In case of USA companies, include the city and the state abbreviation. Unify and apply to the entire document.
Section 2.2.: Describe how soil analyses have been carried out.
Section 2.3.: Describe how precipitation and temperature data have been obtained.
Line 165: Describe how homogenisation has been carried out (grinding, only mixing, etc.).
Section 2.4.: Include calibration curve, r2 and range of linearity for the method (gallic acid).
Section 2.4.: Include conditions of the ultrasonic bath (W, T, ..). Include the same in other sections.
Section 2.5.: Include calibration curve, r2 and range of linearity for the method (catechin).
Section 2.6.: Include calibration curve, r2 and range of linearity for the three methods.
Figure 5 A: Revise C21 and T21 for Zádor.
Minor aspects:
Title: Capitalize each word according the format of the journal.
Lines 25, 26, 138, 238, 244, 247, 248, ….: Put a separation after and before “=”, “<”, “±”. Unify and apply to the entire document.
Lines 122, 133, 163, 195, …..: Capitalize each word according the format of the journal. Unify and apply to the entire document.
Line 127: Define “C” as Control.
Line 128: Define “T” as Treated.
Line 128: Do not put a separation between a number and “%”. Unify and apply to the entire document.
Lines 171, 186, 203,…..: Include “et al.”. Unify and apply to the entire document.
Lines 173, 177, 178, 179,….: Use “mL” and “µL” instead of “ml” and “µl”. Unify and apply to the entire document.
Lines 175, ….: Use “g” or include orbital radius instead of “rpm”.
Line 214: “Trolox equivalent per gram and vitamin E analogue equivalent per gram respectively”?.
Lines 222,….: Put a separation between a number and “ºC”. Unify and apply to the entire document.
Lines 238, 248, 268, 287, 299, 315,….: Put “p” in italics and lower case. Unify and apply to the entire document.
Table 3: Put the numbers in the fotmat “0.798”.
References: The name of the journals must appear abbreviated according to the format of the journal.
Author Response
We would like to thank you for your advices and time for reviewing this paper. We think that your suggestions improved the scientific value of the manuscript. Our notes are added after the advices typed in red.
Major aspects:
Material and Methods: I recommend the authors to include a section of "reagents".
Material and Methods: Lines 164, 174, 176, 177, 231, …: Include the city and country of all the companies cited, and cite the companies of all the reagents and equipment’s employed. In case of USA companies, include the city and the state abbreviation. Unify and apply to the entire document.
Section 2.2.: Describe how soil analyses have been carried out.
Section 2.3.: Describe how precipitation and temperature data have been obtained.
Line 165: Describe how homogenisation has been carried out (grinding, only mixing, etc.)
Section 2.4.: Include calibration curve, r2 and range of linearity for the method (gallic acid).
Section 2.4.: Include conditions of the ultrasonic bath (W, T, ..). Include the same in other sections.
Section 2.5.: Include calibration curve, r2 and range of linearity for the method (catechin).
Section 2.6.: Include calibration curve, r2 and range of linearity for the three methods.
Thank you for your suggestions. New subchapter for presenting reagents was added to the manuscript, also equipment type and parameters were also added as asked. Standard curves, R2 and range of linearity data was added to the manuscript as supplementary material. Soil analysis and weather data collection was added to material and method section.
Figure 5 A: Revise C21 and T21 for Zádor.
Zádor variety for third year was not available for analysis due to crop production issues, thus we did not have results for the third year.
Minor aspects:
Title: Capitalize each word according the format of the journal.
Lines 25, 26, 138, 238, 244, 247, 248, ….: Put a separation after and before “=”, “<”, “±”. Unify and apply to the entire document.
Lines 122, 133, 163, 195, …..: Capitalize each word according the format of the journal. Unify and apply to the entire document.
Line 127: Define “C” as Control.
Line 128: Define “T” as Treated.
Line 128: Do not put a separation between a number and “%”. Unify and apply to the entire document.
Lines 171, 186, 203,…..: Include “et al.”. Unify and apply to the entire document.
Lines 173, 177, 178, 179,….: Use “mL” and “µL” instead of “ml” and “µL”. Unify and apply to the entire document.
Lines 175, ….: Use “g” or include orbital radius instead of “rpm”. ü
Line 214: “Trolox equivalent per gram and vitamin E analogue equivalent per gram respectively”?.
Lines 222,….: Put a separation between a number and “ºC”. Unify and apply to the entire document.
Lines 238, 248, 268, 287, 299, 315,….: Put “p” in italics and lower case. Unify and apply to the entire document.
Table 3: Put the numbers in the fotmat “0.798”.
References: The name of the journals must appear abbreviated according to the format of the journal.
Thank you very much for your insight and suggestions for this paper. Your advices were very helpful during the correction. All aforementioned suggestions were applied to the document.
Round 2
Reviewer 2 Report
I accept the manuscript after major revisions that the authors have correctly made
Author Response
We would like to thank your acceptance and your review.
Reviewer 3 Report
The authors addressed my comments. However, the statistical approach needs to be revised. If they go for a three-way probably they have to reanalyse the data and put all the varieties togeteher in the analyses. As it is presented, it seems that the authors analyses the three factors for the red and white cultivars separately. Probably, this is why no statistical differences were observed in terms of TPC in Figure 2, despite the obvious differences between bars and the low SD values. The same applies for the rest of the Figures.
Then, i suggest to perform a PCA analysis to see if the tested parameters can allow grouping of cultivars.
Author Response
We would like to thank that you found that we addressed your comments. Yes, based on your and the 4th reviewer’s advice, we performed a three-way ANOVA for the overall statistical analysis of the possible influencing factors on the evaluated parameters. In this analysis we have included all the 6 cultivars together, not separating by seed colors. These results are discussed in chapter 3.1. and the results proved statistically that TPC is influenced by all the main factors (cultivar, year and treatment) and the effect of cultivar x treatment interaction was also significant (results are in Table 3). On the other hand, it is also reported in the references that these parameters are influenced by the color of seed, however, in our experiment we found that not the color of seed is the determinant factor, as one of red cultivar has as low TPC and CT content and antioxidant activity as the values of white cultivars. This is why we chosen two-way ANOVA for the further statistical evaluation (as it is discussed in L262-265) and we evaluated the effects of year and N treatment on the different parameters by cultivars separately. In the case of TPC, the year had significant effect for TPC values of all cultivars, while the effect of treatment was significant only in the case of Zádor and Albita. The reason that the different colored grains are in different figures is that this way the difference in red varieties are visually clearly visible and the values and differences for white cultivars are better visible on the different y-axis scale. These findings are also valid for the further parameters and figures.
Thank you for your recommendation for PCA analysis, but the aim of our investigations was the evaluation of effects, but not the classification of cultivars. When we have more results from more years, we will try to get further background information with this method.
Reviewer 4 Report
The authors have made some of the changes suggested but there are still some aspects of the article that need to be improved. Authors should improve their work based on the following comments:
Lines 123, 134, 169, …..: Capitalize each word according the format of the journal. Unify and apply to the entire document.
Lines 147, 148, , ….: Put a separation after and before “=”, “<”, “±”. Unify and apply to the entire document.
Lines 194-196, 226,….: Use “mL” and “µL” instead of “ml” and “µl”. Unify and apply to the entire document.
References: In the name of the journals, after each abbreviated word, use a dot, according the format of the journal.
Author Response
We would like to thank you for your advices and time for reviewing again this paper. Our notes are added after the advices typed in red.
The authors have made some of the changes suggested but there are still some aspects of the article that need to be improved. Authors should improve their work based on the following comments:
Lines 123, 134, 169, …..: Capitalize each word according the format of the journal. Unify and apply to the entire document.
We would like to thank your comment. We missed to change the letters in subtitles – all has been corrected.
Lines 147, 148, , ….: Put a separation after and before “=”, “<”, “±”. Unify and apply to the entire document.
These errors have been corrected – separation space has been added to everywhere.
Lines 194-196, 226,….: Use “mL” and “µL” instead of “ml” and “µl”. Unify and apply to the entire document.
The units have been unified to large L.
References: In the name of the journals, after each abbreviated word, use a dot, according the format of the journal.
All the abbreviated titles have been revised and the missing dots have been added.